# Literary Identification of Differentially Hydroxymethylated DNA Regions for Type 2 Diabetes Mellitus: A Scoping Minireview

**DOI:** 10.3390/ijerph21020177

**Published:** 2024-02-04

**Authors:** Ryan Anh Minh Luong, Weihua Guan, Fue Chee Vue, Jun Dai

**Affiliations:** 1Doctoral Program of Osteopathic Medicine, College of Osteopathic Medicine, Des Moines University, West Des Moines, IA 50266, USA; ryan.a.luong@dmu.edu (R.A.M.L.); fue.c.vue@dmu.edu (F.C.V.); 2Division of Biostatistics & Health Data Science, University of Minnesota School of Public Health, Minneapolis, MN 55414, USA; wguan@umn.edu; 3Department of Public Health, College of Health Sciences, Des Moines University, West Des Moines, IA 50266, USA

**Keywords:** hydroxymethylation, type 2 diabetes mellitus, blood, peripheral, human, twin

## Abstract

Type 2 diabetes mellitus (T2DM) is a public health condition where environmental and genetic factors can intersect through hydroxymethylation. It was unclear which blood DNA regions were hydroxymethylated in human T2DM development. We aimed to identify the regions from the literature as designed in the ongoing Twins Discordant for Incident T2DM Study. A scoping review was performed using Medical Subject Headings (MeSH) and keyword methods to search PubMed for studies published in English and before 1 August 2022, following our registered protocol. The keyword and MeSH methods identified 12 and 3 records separately, and the keyword-identified records included all from the MeSH. Only three case-control studies met the criteria for the full-text review, including one MeSH-identified record. Increased global levels of 5-hydroxymethylated cytosine (5hmC) in T2DM patients versus healthy controls in blood or peripheral blood mononuclear cells were consistently reported (*p* < 0.05 for all). Among candidate DNA regions related to the human *SOCS3*, *SREBF1*, and *TXNIP* genes, only the *SOCS3* gene yielded higher 5hmC levels in T2DM patients with high poly-ADP-ribosylation than participants combined from those with low PARylation and healthy controls (*p* < 0.05). Hydroxymethylation in the *SOCS3*-related region of blood DNA is promising to investigate for its mediation in the influences of environment on incident T2DM.

## 1. Introduction

Type 2 diabetes mellitus (T2DM) is characterized by increased insulin resistance (IR), insulin insensitivity, and blood glucose levels and is becoming increasingly prevalent worldwide [1]. Under normal conditions, blood glucose homeostasis in humans is regulated by the hormones insulin and glucagon, produced from the endocrine pancreas [2]. Insulin reduces the blood glucose level, while glucagon increases the level. Blood glucose levels rise after a meal as dietary carbohydrates are broken down into glucose and enter the bloodstream. In response, pancreatic beta cells release insulin to allow the glucose to enter the body’s cells to provide energy and be stored for later use in the form of glycogen and fat [2]. When blood glucose levels are low, pancreatic alpha cells release glucagon to allow the body to increase glucose levels by producing and reserving glucose [2]. Glucose can be generated from the conversion of stored glycogen into glucose via glycogenolysis and the de novo synthesis of glucose from non-carbohydrate chemicals via gluconeogenesis [2]. Glucose reservation is the process of reducing glucose utilization by inhibiting glycolysis and glycogenesis and enhancing lipolysis and the breakdown of amino acids to provide energy to the body [2]. In T2DM, glucose homeostasis is disrupted. T2DM patients present with increased IR in tissues and dysfunctional insulin secretion by beta cells [1]. Beta cells initially attempt to compensate for the increased IR by producing more insulin. However, this compensatory process increases to unsustainable levels and leads to beta cell burnout, which in turn can decrease insulin production at later stages [1]. The loss of compensatory mechanisms leads to chronic hyperglycemia and T2DM. T2DM can lead to macrovascular diseases like cardiovascular disease and microvascular diseases such as diabetic retinopathy, nephropathy, and neuropathy [3].

The diagnosis of T2DM is pivotal to epidemiological studies of T2DM and clinical settings. Our ongoing twins discordant for incident T2DM (iT2D-dTwin) study included twins from the National Heart, Lung, and Blood Institute (NHLBI) Twin Study (NTS) [4] (NTS iT2D-dTwin Study). In the NTS that was initiated in the late 1960s and had six active follow-up exams until the early 2000s [4], fasting plasma glucose levels and the one-hour plasma glucose levels after a 50-gram glucose load were measured [5]. However, clinical biochemical tests to facilitate a diagnosis of T2DM have been evolving, which have affected and will highly likely affect the population study of T2DM. A better understanding of the current clinical biochemical tests concerning a diagnosis of T2DM is important for population research on T2DM. Clinical biochemical tests relevant to the diagnosis include random plasma glucose tests regardless of fasting status, fasting plasma glucose tests, oral glucose tolerance tests (OGTT), or the hemoglobin A1c (HbA1c) test [6]. The random plasma glucose test measures the current plasma glucose levels at the time of the blood draw. A diagnosis of diabetes requires a random plasma glucose level equal to or greater than 200 mg/dL on two separate occasions or equal to or greater than 200 mg/dL on one symptomatic occasion [7]. The fasting plasma glucose test measures plasma glucose levels after an overnight fast (no caloric intake) for at least 8 h [6]. A fasting plasma glucose level equal to or less than 99 mg/dL is considered normal; between 100 and 125 mg/dL is pre-diabetic; and greater than or equal to 126 mg/dL is diabetic [7]. The OGTT assesses an individual’s response to a fixed oral glucose load. For an OGTT, a patient’s plasma glucose can be measured at fasting and several time points within a specific time interval after the glucose load [8]. The most common OGTT is to measure plasma glucose levels at 2 hours after consuming a glucose load of 75 gram of glucose dissolved in water [6]. A 2-hour plasma glucose level equal to or less than 140 mg/dL after a 75-gram load is normoglycemic, 140–199 mg/dL is pre-diabetic, and greater than or equal to 200 mg/dL is diabetic [7]. In addition, the hemoglobin A1C (HbA1c) test measures the glycated hemoglobin level in the blood and represents the average blood glucose level over the past 2–3 months [7]. The HbA1c concentration has been clinically used to monitor if hyperglycemia has been controlled well over the past 2–3 months in diabetic patients [9]. In the past 15 years, the HbA1c concentration has been considered one of the indicators used to diagnose diabetes [10]. A person with HbA1c % less than 5.7% is considered normal; HbA1c % between 5.7% and 6.4% is considered pre-diabetic; and HbA1c % greater than or equal to 6.5% is considered diabetic [7].

Although it is well known that lifestyle and other environmental factors such as diet, exercise, and smoking contribute to the development of T2DM, the possible underlying pathophysiological mechanisms are not fully elucidated [1]. Epigenetic mechanisms have emerged as novel possible pathophysiological pathways linking environmental factors to the incidence and progression of diseases [1]. Epigenetic modifications refer to changes to the genome without DNA sequence changes [11]. These modifications can alter gene expression via changes in the accessibility of DNA and chromatin. Epigenetic modifications come in various forms, such as histone modification, DNA methylation, and DNA hydroxymethylation [11]. DNA methylation is most commonly studied at the CpG dinucleotide sequence, which serves as a site of transcription initiation and transcription factor interaction [12]. The methylation of 5-cytosine at CpG sites leads to gene silencing. In contrast, DNA demethylation has been less extensively studied. 5-Hydroxymethylcytosine (5hmC) is one product of DNA demethylation, and DNA hydroxymethylation can turn on gene expression [13]. The active conversion of 5-methylcytosines (5mC) to 5-hydroxymethylcytosines (5hmC) occurs via oxidation catalyzed by the ten-eleven-translocation (TET) enzymes [14]. Many studies have focused on the relationship between T2DM and DNA methylation; however, little is known about the relationship between DNA hydroxymethylation and T2DM in humans, which hinders further understanding of the disease process, improvement of screening methods, and reduction of the incidence of the condition.

In our ongoing study of blood DNA hydroxymethylation among twins discordant for incident T2DM (iT2D-dTwin) in the NTS iT2D-dTwin Study, DNA hydroxymethylation was hypothesized as a pathophysiological mechanism linking environmental factors to incident T2DM. Therefore, we aimed to develop a scoping review protocol and identify differentially hydroxymethylated DNA regions related to T2DM from the literature. The DNA regions included non-coding regions, genes, and loci. The literary regions would be the candidate regions to be further investigated for their roles in linking environmental factors to incident T2DM via hydroxymethylation in the ongoing NTS iT2D-dTwin study.

## 2. Materials and Methods

The scoping review protocol to identify differentially hydroxymethylated candidate DNA regions related to T2DM was developed through the adaptation of the Preferred Reporting Items for Systematic Reviews and Meta-Analyses Extension for Scoping Reviews (PRISMA-ScR) [15] and with consideration of common procedures published for the scoping review using PRISMA-ScR guidelines [16,17]. This protocol was registered at https://osf.io/w2rk7/ (accessed on 6 March 2023). The adaptation of PRISMA-ScR was made with the specific intention to use PRISMA’s flow diagram and checklist and consideration of the practical implementation, time constraints, and nature of the scoping review for the ongoing NTS iT2D-dTwin study. The “Funding” section of the PRISMA-ScR checklist table [15] was removed.

### 2.1. Search Strategy for Evidence

PubMed [18] was the only database to be searched for relevant literature. The initial search utilized Medical Subject Headings (MeSH) in PubMed, followed by combinations of free text key terms that covered the NTS iT2D-dTwin Study areas: 5hmC and T2DM. During this process, we included alternative terms and search term variations referencing similar concepts relevant to 5hmC and T2DM, such as 5-hydroxymethylcytosine, 5-methylcytosine/analogs and derivatives, diabetes mellitus, type 2, T2DM, and type 2 diabetes (Appendix A). For this scoping review, we also used optional terms, including but not limited to human, patient, case, and peripheral blood. In addition, other possible or extensive disorders relevant to T2DM included pre-diabetes, hyperglycemia, and high blood glucose. The possible terms might or might not be used for the search, depending on the findings from the search of the required study area. Furthermore, we conducted forward and backward citation tracking and screened literature listed in the “Similar Articles” section listed below the “Abstract” text on the PubMed website interface for each included study. The included studies were limited to the English language and published on PubMed prior to 1 August 2022. This date was selected to satisfy the needs of the ongoing NTS iT2D-dTwin Study.

### 2.2. Evidence Selection Criteria

The published research papers were identified according to the search strategy described above, and the inclusion criteria were as follows:(1)Original research studies that might include quantitative, qualitative, and mixed methods, observational studies, interventions, randomized trials, and experiments.(2)Studies that included type 2 diabetes patients as the primary subjects but might have animals in addition to human subjects.(3)Studies were published on PubMed prior to 1 August 2022, as the initial search was conducted between 1 June and 1 August 2022 for the required concept areas.(4)Studies are written in the English language.(5)Studies that addressed DNA hydroxymethylation and its relation to T2DM.(6)Studies that had comparisons to healthy or non-diabetic reference groups.

As buffy coat DNA samples collected in the NTS [4] were used to measure hydroxymethylation in the NTS iT2D-dTwin Study, the hierarchy of scientific evidence in the reported review was defined as buffy coat DNA > DNA from purified specific blood cells such as peripheral mononuclear cells as well as hydroxymethylation > methylation.

### 2.3. Evidence Assessment

We followed the adapted PRISMA-ScR outlined above and other common procedures [15,16,17] to review each publication. First, duplicate references from independent searches were checked and removed prior to screening. Then, articles were screened by their titles and abstracts for initial eligibility using our primary screening criteria (Section 2.2, Items 1–4). Following this screening, the full-text publications were screened more closely using the selection criteria outlined above (Section 2.2, Items 5–6). Full-text reviews of eligible studies were then conducted by the independent reviewer(s). If questionable or conflicting evidence was identified in an article and the number of articles that passed screening was five or less, reviewers would attempt to contact the corresponding author and/or first author of the publication to resolve concerns. If the reviewer did not receive any responses within one month, the reviewers would discuss the potential exclusion of that publication with the research mentor, senior researcher, or principal investigator, if appropriate.

### 2.4. Reviewer Protocol

Reviewers independently extracted data from the studies that passed the screening and met the selection criteria. For our goal of identifying literature-based candidate DNA regions, we only required one reviewer, but we would ideally like to have at least two reviewers for the completion of the scoping review if possible. If the extracted data did not correspond between reviewers, an additional reviewer would compare the extracted data and attempt to reconcile the evaluated data. If there was only one reviewer, the extracted data comparison and reconciliation would not occur.

If only one reviewer was available or remained due to the unexpected withdrawal of the second independent reviewer during the review process, the following steps would be followed to maintain the integrity of the scoping review: In addition, the limitations of one reviewer must be acknowledged [19].

a.If the second reviewer withdrew after completing a section of the review, the completed portion would be included and compared to that of the other independent reviewer.b.If the second reviewer withdrew prior to completing a section of the review, the work in progress would be excluded, and an additional independent reviewer should be brought in.c.If an additional second reviewer was unable to be obtained within the constraints, the scoping review could be completed based on the results of a single reviewer.

### 2.5. Data Extraction

The details extracted from each publication for the scoping review must report the study information including but not limited to study objective, study population (i.e., participant demographics, sample type, sample size, definition of cases and controls, if applicable, definition of type 2 diabetes), year of data collection, study design (i.e., case-control, nested case-control, cross-sectional, and cohort studies, intervention, randomized controlled trials with or without blinding, etc.), inclusion and exclusion criteria, methods for hydroxymethylation measurement, candidate DNA loci/genes/regions, types of DNA samples (buffy coat DNA, DNA from peripheral mononuclear cells or other cells, etc.), study setting, statistical analysis with the specification of multiple-testing adjustment, study limitations, bibliographic information (i.e., title, author(s), year of publication, journal, etc.) and results. The extracted results included differentially hydroxymethylated loci/genes/DNA regions; the magnitude of differential hydroxymethylation compared to the reference group or associated with the outcome; the direction of differential hydroxymethylation compared to the reference group or associated with the outcome (i.e., hyper-hydroxymethylation or hypo-hydroxymethylation in diabetic patients relative to non-diabetic patients); and a multiple-testing adjusted or nominal *p*-value. If differentially hydroxymethylated loci/genes/DNA regions were not studied or reported but global changes in DNA hydroxymethylation were reported, this situation should be clearly noted.

### 2.6. Summarizing and Reporting

The extracted details, as described in Section 2.5, were summarized in a table. Information concerning the consistency of findings and infrequent findings across studies was evaluated, reported, and discussed. Quantitative analysis was performed when possible and appropriate. All the extracted studies were not assessed for methodological quality. Instead, the research limitations and advantages of the extracted studies were assessed to provide insight into existing evidence and gaps in knowledge and/or research that could be investigated in future research studies.

### 2.7. Ethics and Dissemination

Although ethics approval for the scoping review was not required, the scoping review was part of an ongoing study approved by the Institutional Review Board (IRB) of Des Moines University (IRB-2020-39, 25 November 2020).

## 3. Results

Figure 1 illustrates the literature search process. A total of 12 records were identified through the PubMed search using keywords (Table 1), of which three records were also identified using MeSH terms. After all, 12 records were screened based on the abstract, and 9 sources were removed following our inclusion criteria. Therefore, a total of three full-text articles were assessed and included in the scoping review: Pinzon-Cortes et al. [20], Yuan et al. [21], and Zampieri et al. [22]. The evidence was narratively synthesized because of the very limited article number.

Pinzon-Cortes et al. [20] investigated differences in global DNA hydroxymethylation in the peripheral blood cells between 44 T2DM patients (with further stratification into 19 well-controlled T2DM patients with HbA1c < 7% and 25 poorly-controlled T2DM patients with HbA1c ≥ 7%) and 35 healthy non-T2DM control patients. Global 5hmC levels measured in ng and percentage were reported. The statistically significant level and information about a one-tailed or two-tailed statistical test were not found in the article. Researchers graphically illustrated percentages of global 5hmC in total DNA using 2-dimensional coordinates for group comparisons. Pinzon-Cortes et al. reported a significantly high global 5-hydroxymethylcytosine (5hmC) percentage in blood cells of T2DM patients vs. controls (*p* = 0.045), implying that a statistical significance level of 0.05 (i.e., alpha = 0.05) was used. Additional findings showed that differences in global 5hmC percentages between well-controlled T2DM patients (mean ± unspecified dispersion index, 0.09%  ±  0.21%) and control patients (0.61%  ±  1.03%) were not statistically significant. When the global 5hmC percentage of poorly-controlled T2DM patients was compared to that of well-controlled T2DM patients, poorly-controlled T2DM patients had significantly increased global 5hmC levels compared to those of well-controlled T2DM patients (0.20%  ±  0.53% vs. 0.09%  ±  0.21%, *p* = 0.015) as well as compared to those of healthy controls (0.20%  ±  0.53% vs. 0.61%  ±  1.03%, *p* = 0.002), and poorly-controlled T2DM patients had significantly increased global 5hmC levels compared to those of a combined well-controlled T2DM and healthy control (*p* = 0.0034). The global 5hmC levels in ng were 0.21  ±  0.53, 0.08  ±  0.15, and 0.76  ±  1.28 for poorly controlled patients, well-controlled patients, and healthy controls, respectively. However, no results for group comparisons of global 5hmC levels in ng were reported.

Yuan et al. [21] investigated differences in global DNA hydroxymethylation measured as a molar ratio of 5hmC to deoxycytosine (dC) in the whole blood cells in an age- and gender-matched case-control study between 104 T2DM patients and 108 healthy control patients and in an additional animal case-control study comparing six diabetic and six non-diabetic male Wistar rats. In the “Statistical Methods” section, *p* < 0.05 was stated to be statistically significant without information about a one-tailed or two-tailed statistical test. Major findings from this study showed significantly increased levels of global DNA 5hmC in whole blood cells of T2DM patients compared to those of control patients (0.0233%  ±  0.0169% vs. 0.0194%  ±  0.0075%, *p* = 0.036). In addition, T2DM patients were found to have significantly increased fasting blood glucose levels (*p* <  0.001), and there was a significantly positive association between fasting blood glucose levels and 5hmC levels after adjusting for age and gender (*β*  =  0.224, *p* =  0.0065). When T2DM patients were further stratified into T2DM patients with (MD+) or without (MD−) macrovascular diseases (MD), differences between T2DM MD− patients and controls or between T2DM MD+ and T2DM MD− groups were not statistically significant. By contrast, differences in 5hmC levels in T2DM MD+ patients and controls were significantly different (*p* = 0.020). In rat model testing, diabetic rats had marginally statistically significant elevated levels of global DNA hydroxymethylation compared to control rats (*p* = 0.054).

Zampieri et al. [22] investigated differences in PARylation levels (PAR) and DNA hydroxymethylation status in peripheral blood mononuclear cells between 61 T2DM subjects (who were further stratified into low PAR and high PAR subjects) vs. 48 controls in an age- and gender-matched case-control study. Poly(ADP)-ribosylation (PARylation) was a post-translational protein modification process catalyzed by the poly(ADP-ribose) polymerase (PARP) enzymes and played a role in DNA damage and repair [32]. As illustrated in a previously published review by this research group [33], PARylation controls DNA methylation patterns. Levels of global DNA hydroxymethylation were measured as the percentage of 5hmC in genomic DNA. In addition, this study also analyzed differences in hydroxymethylation for the *SOCS3* (suppressor of cytokine signaling 3), *SREBF1* (sterol regulatory element binding transcription factor 1)*,* and *TXNIP* (thioredoxin interacting protein) genes among 10 low PAR patients, 10 high PAR patients, and 10 controls. Using the EpiTYPER assay and DNA immunoprecipitation (mDIP), this study analyzed the following DNA regions and their respective genes: *SOCS3*: chr 17 bp 76,354,370–76,354,900; *SRBF1*: chr 17 bp 76,354,482–76,354,692; and *TXNIP*: chr 1 bp 145,441,299–145,441,885. Zampieri et al. reported a strong positive correlation of PAR with HbA1c. Researchers explicitly reported the mean and standard deviation (SD) of global 5hmC levels for study groups and stated *p* ≤ 0.05 to be statistical significance in Table 1, and revealed significantly higher global 5hmC levels in T2DM patients (mean  ±  SD, 0.037% ± 0.009%) compared to controls (0.070%  ±  0.037%) (*p* < 0.001). By contrast, for gene-specific 5hmC, using 2-dimensional coordinates, researchers graphically illustrated percentage 5hmC levels for the three-group comparison. The analyses of mDIP enrichment data showed that there were significantly increased 5hmC levels in *SOCS3* in high PAR T2DM patients vs. low PAR T2DM and controls individually (Dunn-Bonferroni test *p* ≤ 0.05 for both), but there were no significant differences in 5hmC levels found between groups for the *SREBF1* and *TXNIP* genes. Furthermore, the analysis showed that there were significantly increased 5hmC levels in *SOCS3* in high PAR T2DM patients vs. a combined low PAR T2DM and control patient group (Dunn-Bonferroni test *p* ≤ 0.01). For the specific CpG site in gene *SOCS3,* the analyses of methylation-sensitive PCR data showed significantly high 5hmC levels in *SOCS3* CpG 16 in high PAR T2DM patients vs. low PAR T2DM and controls separately (Dunn-Bonferroni test *p* ≤ 0.01 for both).

## 4. Discussion

The purpose of this scoping review was to identify DNA regions hydroxymethylated differentially by T2DM from the literature as designed in the ongoing NTS iT2D-dTwin Study. We did not find any record concerning incident T2DM. Instead, only three case-control studies met our search and screening criteria for the full-text review, of which one study included an additional animal experiment. The keyword method identified all three studies, but the MeSH search identified only one. Therefore, we found that the keyword search proved to be the preferable method for our review. All three studies consistently found increased global levels of 5-hydroxymethylated cytosine in blood or peripheral blood mononuclear cells among T2DM patients versus healthy controls. Given the lack of statistical significance for differential hydroxymethylation in the study by Zampieri et al. [22] and the relative small sample size in our ongoing study, without consideration of the pathophysiological role of *SREBF1* and *TXNIP* genes in IR and T2DM [34,35], we identified the DNA region related to the *SOCS3* gene as a promising primary candidate DNA region for future investigation of its hydroxymethylation in blood associated with incident T2DM.

The potential pathophysiological mechanisms linking hydroxymethylation of DNA regions/genes/loci to the development of IR and T2DM could be inferred from prior methylation research and the function and expression of their mapped genes. The *SOCS3* expression inhibited insulin signaling and thus contributed to the development of IR and T2DM [36]. Low levels of methylation of the *SOCS3* gene reversed gene silencing of 5-cytosine methylation. Hypomethylation of the *SOCS3* gene was functionally equivalent to hyper-hydroxymethylation in the way that both turned on the gene and thus resulted in an increased expression of *SOCS3*. Hypomethylation of the *SOCS3* gene was related to inflammation-induced IR [37], and so was hypermethylation of the *SOCS3* gene. Circulating interleukin-6 (IL-6) as a biomarker for systemic low-grade inflammation concerning environmental factors/determinants and cardiovascular diseases in prior twin studies [38,39,40]. Tumor necrosis factor alpha (TNFα) and interleukin-1β (IL-1β) were pro-inflammatory cytokines that stimulated the synthesis and secretion of IL-6. IL-6 was also related to IR and T2DM by inducing the expression of SOCS3 proteins [36], which in turn disrupted the binding between the insulin receptor and insulin receptor substrate-1 (IRS-1) and degraded IRS-1 to inhibit insulin signaling [36,41].

Although we did not list that *SREBF1* and *TXNIP* genes were promising to be our primary candidate DNA regions in our ongoing study, the pathophysiological role of *SREBF1* and *TXNIP* genes in IR and T2DM was discussed here to provide comprehensive knowledge as their hydroxymethylation related to T2DM was investigated by Zampieri et al. [22]. Zampieri et al. were inspired to study *SOCS3*, *SREBF1*, and *TXNIP* genes [22] since their methylation was associated with the onset of T2DM in a prior whole blood epigenome-wide study of Indian Asian participants published by Chambers et al. in 2015 [42]. We noticed that methylation of specific CpG sites cg18181703 (*SOCS3*), cg11024682 (*SREBF1*), and cg19693031 (*TXNIP*) in blood DNA was not associated with future T2DM risk in the Botnia prospective study of 148 1:1 nested case-control pairs for incident T2DM in Finland, including 11 male and 8 female monozygotic twin pairs discordant for T2DM, as described in the abstract of the paper [43]. Unfortunately, Dayeh et al. did not separately report results from discordant twin pairs [43].

The *SREBF1* gene encoded the sterol regulatory element-binding proteins (SREBP) SREBP-1a and SREBP-1c as transcription factors [35]. Both proteins regulate fatty acid and cholesterol synthesis. Low levels of SREBP-1c might be related to the development of IR, as SREBP-1c expression was reduced in the skeletal muscle and adipose tissue of patients with T2DM [35]. SREBP-1 was needed in macrophage-mediated resolution of pro-inflammatory Toll-like receptor 4 (TLR4) signaling by the programming of anti-inflammatory fatty acid biosynthesis and the uncoupling of NFκB binding from gene activation [44]. TLR4 signaling could induce interferons-1 and TNFα [44]. TNFα stimulates IL-6 production, contributing to IR and T2DM [36].

The *TXNIP* gene encodes thioredoxin-interacting protein (TXNIP). Thioredoxin is an antioxidant protein. Interaction between TXNIP protein and thioredoxin increased oxidative stress through inhibiting antioxidant protein thioredoxin, initiated inflammation by activating inflammatory signaling via the nucleotide-binding oligomerization domain (NOD)-like receptor protein-3 (NLRP3) inflammasome, induced mitochondrial stress-induced apoptosis, and stimulated inflammatory cell death (pyroptosis) [34]. The NLRP3 inflammasome led to the generation of IL-1β and interleukin-18, which caused pyrotosis [34]. TXNIP impacted β-cell survival, function, and glucose homeostasis and was upregulated in T2DM [34]. TXNIP-mediated redox inhibition was associated with the premature death of insulin-secreting cells in patients with diabetes [34]. The inhibition of TXNIP enhanced glucose tolerance and insulin sensitivity versus wild-type mice after eight weeks of a high-fat diet experiment [34]. Mice gene Txnip downregulation inhibited β-cell apoptosis and preserved β-cell mass in mice and thus prevented the mice from obesity-induced diabetes [45]. As illustrated by Chambers et al., *TXNIP* was one of the most glucose-responsive genes expressed in human islets and was essential in cellular redox-oxidative stress management, pancreatic β-cell functioning maintenance, and glucose sensing [42].

All three records consistently reported increased circulating global levels of 5hmC in T2DM patients versus healthy controls [20,21,22]. Of significant note, in Zampieri et al., when comparing global levels of 5hmC in T2DM patients versus healthy controls and excluding data from alcohol and tobacco consumers, global levels of 5hmC in T2DM patients versus healthy controls were not significantly different [22]. However, it was noted that the shift from previously significant comparisons to non-significant comparisons could be due to the decrease in sample size. In Pinzon-Cortes et al. [20], we found discrepancies in the data regarding 5hmC levels reported in their Table 1 and Figure 1 and communicated with the corresponding author for clarification. The corresponding author confirmed that an error was made in Table 1, but that the data reported in Figure 1 and in the results section were correct.

We identified one candidate DNA region to be further analyzed: the DNA region related to *SOCS3*, as one source reported significant differences in 5hmC levels in this region between T2DM and healthy control groups [22]. Although no differences in 5hmC levels were reported in the DNA regions related to *SREBF1* and *TXNIP*, given the limited study sample size, composition, and location effects, the 5hmC quantification and analysis could be repeated. Given the limited study availability relevant to our area of study, alternative search criteria should be considered. Other potential candidate DNA regions might be identified based on existing published data reporting DNA regions with differences in DNA methylation between T2DM and control groups.

One limitation to note was that there were no 5hmC levels measured or reported at the DNA region/gene level for *SOCS3* in the overall T2DM group prior to stratification [22]. Attempts to contact the lead author and corresponding author [22] to request supplemental data were made but were unsuccessful in receiving a response. Another limitation was that the dispersion index was not explicitly specified in an article [20]. In addition, one limitation common to all three articles was the lack of an explicit statement of the one-tailed or two-tailed statistical test. The limitation of our reported review was that only one reviewer completed the entire scoping review; there was no cross-checking opportunity between two independent reviewers. However, since it was very less likely to miss the high-impact findings, our review met the goal for our ongoing NTS iT2D-dTwin Study.

## 5. Conclusions

Using our registered scoping review protocol to search PubMed, we found that the human study of circulating hydroxymethylation was rare for incident T2DM with no records in PubMed and very limited for the presence of T2DM with 3 out of 12 records for the final full-text review. The keyword search method identified more records than the MeSH method for our study. The three publications for the full-text review were from three different case-control studies and consistently reported that T2DM patients had significantly increased global levels of 5-hydroxymethylated cytosine in blood or peripheral blood mononuclear cells when compared to healthy controls. The DNA region related to the *SOCS3* gene is a promising primary candidate DNA region for future investigation of its hydroxymethylation in blood as a potential pathophysiological pathway linking the environment to incident T2DM.

## Figures and Tables

**Figure 1 ijerph-21-00177-f001:**
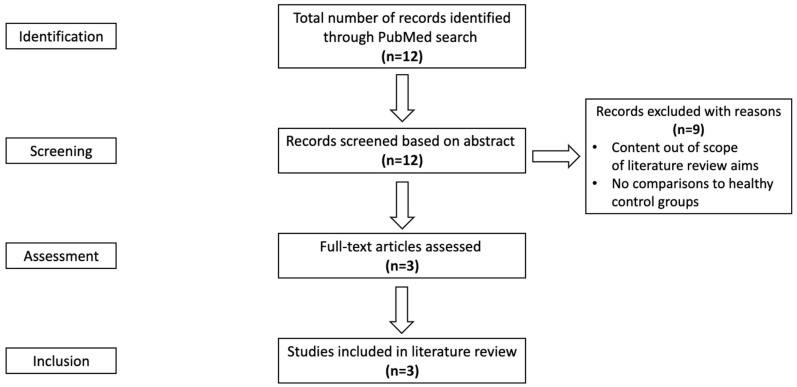
Study selection: PRISMA-scoping review flow diagram.

**Table 1 ijerph-21-00177-t001:** PubMed search results.

Search	Source Title	Author
MeSH	Alterations of 5-hydroxymethylcytosines in circulating cell-free DNA reflect retinopathy in type 2 diabetes.	Han et al. [23]
	5-Hydroxymethylcytosines in Circulating Cell-Free DNA Reveal Vascular Complications of Type 2 Diabetes.	Yang et al. [24]
	Hyperglycemia affects global 5-methylcytosine and 5-hydroxymethylcytosine in blood genomic DNA through upregulation of SIRT6 and TETs. **	Yuan et al. [21]
PubMed Keyword search	Alterations of 5-hydroxymethylcytosines in circulating cell-free DNA reflect retinopathy in type 2 diabetes.	Han et al. [23]
	Increased PARylation impacts the DNA methylation process in type 2 diabetes mellitus. *	Zampieri et al. [22]
	5-Hydroxymethylcytosines in Circulating Cell-Free DNA Reveal Vascular Complications of Type 2 Diabetes.	Yang et al. [24]
	Effect of diabetes status and hyperglycemia on global DNA methylation and hydroxymethylation. *	Pinzon-Cortes et al. [20]
	Hyperglycemia affects global 5-methylcytosine and 5-hydroxymethylcytosine in blood genomic DNA through upregulation of SIRT6 and TETs. **	Yuan et al. [21]
	DNA modifications: function and applications in normal and disease States.	Liyanage et al. [25]
	5-Hydroxymethylcytosine Remodeling Precedes Lineage Specification during Differentiation of Human CD4(+) T Cells.	Nestor et al. [26]
	Genome-wide Analysis Reflects Novel 5-Hydroxymethylcytosines Implicated in Diabetic Nephropathy and the Biomarker Potential.	Yang et al. [27]
	Epigenetic Modifications in the Biology of Nonalcoholic Fatty Liver Disease: The Role of DNA Hydroxymethylation and TET Proteins.	Pirola et al. [28]
	Machine-learning to stratify diabetic patients using novel cardiac biomarkers and integrative genomics.	Hathaway et al. [29]
	Apolipoprotein E4 and Insulin Resistance Interact to Impair Cognition and Alter the Epigenome and Metabolome.	Johnson et al. [30]
	Stable Oxidative Cytosine Modifications Accumulate in Cardiac Mesenchymal Cells From Type 2 Diabetes Patients: Rescue by α-Ketoglutarate and TET-TDG Functional Reactivation.	Spallota et al. [31]

* Studies were included in the full-text and narrative review. ** Studies included in the full-text and narrative review were identified with both the keyword and MeSH methods.

## Data Availability

Not applicable.

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
