# Peer review of "Literary Identification of Differentially Hydroxymethylated DNA Regions for Type 2 Diabetes Mellitus: A Scoping Minireview"

_ijerph, 2024, doi:10.3390/ijerph21020177_

Round 1

Reviewer 1 Report

Comments and Suggestions for Authors

the review by Luong et al describes the effect of DNA hydroxymethylation of the development of T2D. Based  on the literature review in pubmed they conclude that  SOCS3 gene hydroxymethylation may associated with T2D induction. The study is fine. I suggest the followings

1- Since the review is relatively short I suggest the title to be '' Literary Identification of Differentially Hydroxymethylated DNA Regions for Type 2 Diabetes Mellitus: A Scoping Minireview.

2- I suggest the authors to discuss how the hydroxymethylation of the  SOCS3 gene may induce insulin insensitivity and T2D.

3- Why the authors only search in pubmed and not ISI web of knowledge or scopus.  

4- the authors should discussed more from pathophysiological point of view why the hydroxymethylation of the SREBF1, and TXNIP genes are excluded  from their review as epigenetic factors that may implicated in insulin resistance and T2D

Comments on the Quality of English Language

Minor English revision may be required. 

Author Response

Dear Reviewer 1,

Reviewer 2 Report

Comments and Suggestions for Authors

Luong et al. describe a scoping review of 3 studies to identify differentially hydroxymethylated DNA regions for Type 2 Diabetes. While the low number of available studies reduces the impact of the review, overall, the review is conducted to appropriate standards, and (generally) both methods and results are well described. However, I have some minor things I would like to see addressed before publication of this manuscript (mostly in the order they appear in the manuscript):

·        Line 39: tab instead of space between end of sentence and start of new one. Should this be a space? Or was this intended as the start of a new paragraph (which given the context would make sense)? Or does this only appear to be a tab because of the line indentation?

·        Lines 52-75: This paragraph includes an in-depth description of clinical biochemical tests to diagnose T2D. While some indication of how T2D is commonly diagnosed is relevant in the general introduction of any review on T2D, I wonder about the relevance of including such a thorough description in this particular review.

· Line 80: tab instead of space between end of sentence and start of new one. Should this be a space? Or was this intended as the start of a new paragraph (which given the context would not make sense)? Or does this only appear to be a tab because of the line indentation?

·        Initially I was pleased to see that the protocol for the scoping review was registered with OSF (line 111), however, the protocol has not been made publicly available yet, thus I could not access the protocol to assess whether it matches what is described in the manuscript (unless I would request and receive access but that would make this a non-anonymous review which does not have my preferences and I, therefore, opted against). Having this kind of registration publicly available in the review stage of a manuscript would make more sense. Still, the editor should check this registration is publicly available at the time of publication at least. Moreover, without having the protocol available, it is hard to assess the exact nature or reasoning behind the changes made to the PRISMA-ScR.

·        When I open the supplementary file, I see a copy of the manuscript, not the expected Table S1; likely an uploading error but needs to be corrected.

·        It would be good to add to Table 1 an indicator of the studies included in the narrative review.

· Line 263: effect size reported as a beta coefficient, while the sentence on 262 suggests correlation analyses were performed, if true would reporting the effect as r not be more accurate than reporting B? If results were reported as a beta coefficient, suggest replacing correlation on 262 with association.

· Line 270: PARylation is a term readers may be unfamiliar with, to understand the results of Zampieri study it would help readers if a definition/explanation of PARylation is provided.

·        line 290: SOCS3 not italicized.

·        line 316 extra space between an & error.

·        line 319: SOCS = SOCS3?

Author Response

Dear Reviewer 2,
